# A Self-Supervised Method for Mapping Human Instructions to Robot Policies

## Abstract

In this paper, we propose a modular approach which separates the instruction-to-action mapping procedure into two separate stages. The two stages are bridged via an intermediate representation called a *goal*. The first stage maps an input instruction to a goal, while the second stage maps the goal to an appropriate policy selected from a set of robot policies. The policy is selected with an aim to guide the robot to reach the goal as close as possible. We implement the above two stages as a framework consisting of two distinct modules: an instruction-goal mapping module and a goal-policy mapping module. Given a human instruction in the evaluation phase, the instruction-goal mapping module first translates the instruction to a robot-interpretable goal. Once a goal is derived by the instruction-goal mapping module, the goal-policy mapping module then follows up to search through the goal-policy pairs to look for policy to be mapped by the instruction. Our experimental results show that the proposed method is able to learn an effective instruction-to-action mapping procedure in an environment with a given instruction set more efficiently than the baselines. In addition to the impressive data-efficiency, the results also show that our method can be adapted to a new instruction set and a new robot action space much faster than the baselines. The evidence suggests that our modular approach does lead to better adaptability and efficiency.

## 1 Introduction

Understanding human instructions and interpreting them into actions have long been a crucial need and research focus for autonomous robots (Winograd, 1972; Thomason et al., 2017). As autonomous robots are increasingly prevalent and capable of performing versatile skills nowadays (Antunes et al., 2016; Guadarrama et al., 2013; Karamcheti et al., 2017; Spranger et al., 2014), developing an efficient way that enables them to quickly satisfy such a need has also become essential and necessary. Traditionally, this is treated by researchers as a semantic-parsing and instruction-to-action mapping problem. A number of earlier approaches Chen & Mooney (2011); Matuszek et al. (2010; 2013); Mei et al. (2016) propose to learn a semantic parser that maps natural languages or human instructions to sequences of actions executable by a robot in a supervised fashion. The technique of using a probabilistic graphical model for inferring the corresponding robot action given a human instruction has also been investigated in Tellex et al. (2011). Although these approaches have been well explored and mostly validated in their specific problem domains (e.g., navigation) (Mei et al., 2016), they typically require a large amount of human supervision (e.g., data annotation) and/or linguistic knowledge (e.g., the syntax of instructions). In case that the annotated action sequences of a robot or the relevant linguistic background is not directly available, however, developing such an instruction-to-action mapping procedure becomes no longer straightforward.

Researchers in recent years attempt to relax the above constraints by adopting two alternative approaches: reinforcement learning (RL) and imitation learning (IL). Both approaches seek to learn a behavior policy $\pi$ for an agent that executes the given instructions appropriately. The authors in Branavan et al. (2009) employ an RL learner to map text instructions in documents to computer actions. A reward shaping technique is presented in Misra et al. (2017) to train an RL agent to jointly reason about linguistic and visual inputs in a fully observable simulated block world. In Hermann et al. (2017), an RL agent is trained to follow the provided human instructions by learning a joint representation of the perceived image and the text command. Similar to Hermann et al. (2017), the authors in Chaplot et al. (2017) propose to incorporate a multimodal fusion unit, called *gated-*

*attention*, for combining the representations of the instruction and the image. On the other hand, several IL-based techniques have also been leveraged in autonomous robots for executing a diverse range of instructions in a number of tasks (Argall et al., 2009). Although these RL- and IL-based approaches enable a robot to learn effective $\pi$'s to deal with the instruction-to-action mapping problem with little human supervision, they usually train their $\pi$'s in an end-to-end manner. This prevents $\pi$ from easy adaption to unfamiliar sets of instructions or new robots with different action spaces, unless dedicated training data are provided additionally.

To deal with the issues mentioned above, we propose a modular approach which separates the instruction-to-action mapping procedure into two separate stages. The two stages are bridged via an intermediate representation called a *goal*. The first stage maps an input instruction $c$ to a goal $g$, while the second stage maps $g$ to an appropriate policy $\pi$ selected from a set of robot policies $\Pi$. The policy $\pi$ is selected with an aim to guide the robot to reach $g$ as close as possible. Different from the previous end-to-end training methods, these two stages are trained separately as long as they both agree on the same goal representation. The modular approach allows either of the two stages to be replaced by another implementation. For instance, replacing the first stage by another mapping function ($c' \Rightarrow g$) allows a different set of instructions $c'$ (e.g., different languages or different instruction syntaxes) to be used to control the robot. On the other hand, replacing the second stage enables maneuvering a different robot with an action space distinct from the original one under the same instruction set. The modular nature of our approach enables a robot to adapt to a new instruction set quickly, and allows different robots to be maneuvered by the same instruction-to-goal mapping function, as long as the goal representation is identical.

In this paper, we implement the above two stages as a framework consisting of two distinct modules: an instruction-goal mapping module and a goal-policy mapping module. The first module embraces the concept of metric learning, which aims to train a metric function called the *distance function* as a non-linear regressor to evaluate whether a given instruction $c$ is close to a goal $g$ or not. The second module utilizes the concept of goal exploration process (GEP) (Forestier et al., 2017) to discover in advance a diversified set of policies in $\Pi$ in an unsupervised manner. It keeps expanding $\Pi$ during the training phase to enhance the capability of the robot to reach a wide range of goals. The goals and their corresponding policies are indexed as goal-policy pairs, and are stored in a buffer contained in the second module. Given a human instruction $c$ in the evaluation phase, the instruction-goal mapping module first translates $c$ to a robot-interpretable goal $g$. The translation is performed by using the distance function output as a metric to search for the most similar $g$ in the goal space $G$. The implementation of the goal-searching procedure is dependent on the nature of the goal. For instance, a straightforward enumeration algorithm is likely an appropriate option for a discrete goal, while a more complicated sampling-based search scheme is required for a goal in the continuous space. Once a goal $g$ is derived by the instruction-goal mapping module, the goal-policy mapping module then follows up to search through the goal-policy pairs to look for a pair containing the most similar goal to $g$. The selected pair is then retrieved from the buffer, and used to extract a $\pi$ to serve as the policy to be mapped by $c$.

To demonstrate the strengths and efficiency of our modular approach, we compare the performance of our method with a number of baselines in two types of benchmark tasks: goal-oriented and trajectory-oriented tasks. The former type simply requires the robot to approach a specific goal, while the latter type demands the robot to accomplish the assigned task in the way specified by the instruction. In our experiments, these two types of tasks are both implemented in a simple 2-dimensional points of mass environment as well as a more challenging locomotion control task simulated by the MuJoCo physics engine (Todorov et al., 2012). The baselines are adapted from the RL- and IL- based approaches presented in Dhariwal et al. (2017); Bain & Sommut (1999). Our experimental results show that the proposed method is able to learn an effective instruction-to-action mapping procedure in an environment with a given instruction set more efficiently than the baselines. Compared to the RL baseline, our method is able to achieve a comparable or even superior performance by fewer interactions with the environment. Moreover, our approach consumes less human demonstration data than the IL baseline. In addition to the impressive data-efficiency, the results also show that our method can be adapted to a new instruction set and a new robot action space much faster than the baselines. The above evidence suggests that our modular approach does lead to better adaptability and efficiency. The main contributions of the paper are summarized as the following:

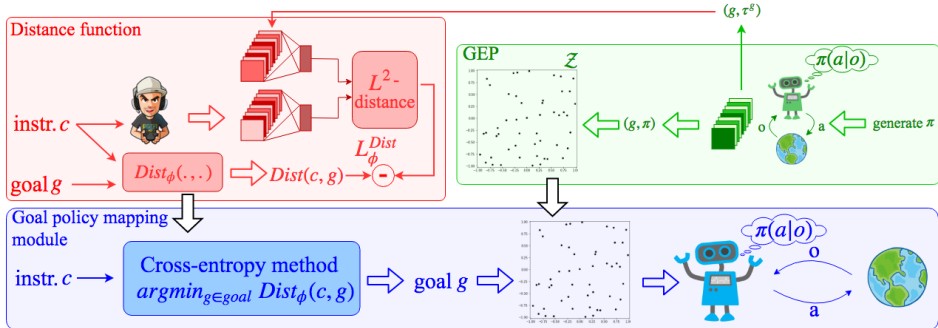

**Figure 1:** **Overview of the proposed framework.**

- A modular framework for mapping human instructions to robot policies via an intermediate goal representation.
- A unsupervised training method that requires neither action labels of the robot nor instruction-goal pairs.
- A distance function for evaluating the similarity between a human instruction and a goal.

The rest of this paper is organized as follows. Section 2 introduces background material. Section 3 walks through the proposed methodology, its implementation details, and the training procedure. Section 4 presents the experimental results and the ablation study. Section 5 concludes this paper.

## 2 BACKGROUND

In this section, we introduce background material. We first provide an overview of multi-goal RL. We then discuss the concepts and the relevant research works on metric learning.

### 2.1 GOAL EXPLORATION PROCESS

Goal exploration process (GEP) is an unsupervised policy search algorithm proposed in Forestier et al. (2017) that aims to discover a populations of policies $\pi$'s corresponding to the goals $g$'s. In the framework of GEP, the goal $g$ is often a behavioral feature characterizing the trajectory produced by performing the corresponding policy $\pi$ in the environment. GEP is composed of two stages: a bootstrap stage and a random exploration. The former first samples sets of policies from $\Pi$ and observes the goals $g$'s resulted from the policies $\pi$'s, while the latter samples a goal $g$ from $G$ which the algorithm looks for policy $\pi$ that able to reach the specific goal. In both stages, whenever the new goal is observed, the corresponding goal-policy pair is stored. To ensure the goal $g$ is novel to the previous ones, GEP adds random noises to perturb the paired policies. GEP explores $G$ in an unsupervised fashion.

### 2.2 METRIC LEARNING

The primary goal of metric learning is to learn a metric function for measuring the relationship between different sets of data points. Theses sets may come from the same data domain or different ones. Metric learning has recently received great attention from researchers due to its applicability to a wide variety of research domains, such as RL (Taylor et al., 2011), partition problems (Lajugie et al., 2014), information retrieval (Lebanon, 2006; Lee et al., 2008; McFee & Lanckriet, 2010), etc. The metric functions in the previous works are usually implemented in the form of distance functions. However, the concept of metric learning is generalizable to logical relations among data points. As a metric function is commonly trained with limited amount of training data, it is important for it to learn the general relations between its input data sets. For instance, the distance function for measuring the similarity of faces presented in Chopra et al. (2005) predicts a small value when the inputs to it come from the same person, and a large value when the inputs are from different people. Metric learning has also been used in RL to filter out features irrelevant for decision making (Taylor et al., 2011). In this paper, we apply the concepts of metric learning for mapping human instructions to robot skills, which have yet been explored in the previous literature.

## 3 PROPOSED METHODOLOGY

In this section, we present the framework and the implementation details of the proposed approach. We first provide an overview of the framework, followed by an in-depth explanation of the modules.

### 3.1 OVERVIEW OF THE PROPOSED FRAMEWORK

Fig. 1 illustrates the proposed framework, which targets at mapping a human instruction code $c \in C$ to a behavior policy $\pi \in \Pi$ of a robot, where $C$ and $\Pi$ represent the instruction space and the set of policies owned by the robot, respectively. The framework consists of two modules: an instruction-goal mapping module and a goal-policy mapping module. The former first maps a given instruction $c$ to a goal $g \in G$ (where $G$ is the goal space), while the latter maps $g$ to an appropriate policy $\pi$ that leads the robot to another $g' \in G$ as close to $g$ as possible. The main objective of the instruction-goal mapping module is to learn a mapping function between the two spaces $C$ and $G$. On the other hand, the goal-policy mapping module aims to continuously extend $\Pi$ with new and diversified policies by exploration such that the chances of finding a better $\pi$ with a lower distance between $g$ and $g'$ is increased. These two modules are trained concurrently during the training phase.

The instruction-goal mapping module of the proposed framework comprises two distinct components: a trajectory encoder $f$ and a distance function $Dist_\phi(c, g)$ parameterized by a set of trainable parameters $\phi$. The primary function of the trajectory encoder is to extract a representation $h$ from a given trajectory $\tau$, which is composed of a sequence of observations $o's$ defined as follows:

$$\tau = (o_0, o_1, \cdots, o_t, o_{t+1}, \cdots, o_T),  \tag{1}$$

where $t$ represents the timestep in $\tau$, and $T$ the horizon of $\tau$. In this paper, the trajectories corresponding to human demonstration (according to $c$) and robot's real path (ended at $g$) are denoted as $\tau^c$ and $\tau^g$, respectively. The architecture of the trajectory encoder $f$ is defined and explained in detail in Section 3.2. The representation $h$ is related to $\tau$ and is used to train $Dist_\phi(c, g)$ to evaluate the similarity between any given pairs of $c$ and $g$. A number of $\tau^c$'s and $\tau^g$'s corresponding to a diverse range of $c$'s and $g$'s are prepared in advance as the training data for $Dist_\phi(c, g)$. The trajectories $\tau^c$'s are collected by human demonstrations, while the trajectories $\tau^g$'s are collected by performing various robot policies in $\Pi$, as depicted in Fig. 1. Instead of directly comparing each pair of $\tau^c$ and $\tau^g$, the trajectory encoder $f$ is used to encode each $\tau^c$ and $\tau^g$ into $h_{\tau^c}$ and $h_{\tau^g}$, respectively. As $h_{\tau^c}$ and $h_{\tau^g}$ serve as the representations of $\tau^c$ and $\tau^g$, the Euclidean distance ($L^2$ distance) between these two representations is thus employed as a measure of the distance between $\tau^c$ and $\tau^g$. We train $Dist_\phi(c, g)$ as a regressor to estimate the difference between arbitrary pairs of $h_{\tau^c}$ and $h_{\tau^g}$. The trained $Dist_\phi(c, g)$ is then used during evaluation for mapping $c$ to $g$.

The goal-policy mapping module of the framework consists of two separate processes: a goal-exploration process (GEP) and a goal-searching process (GSP). These two processes are illustrated in Fig. 1. GEP is developed based on the architecture of Forestier et al. (2017), and is described in detail in Section 3.3.1. During the training phase, GEP extensively explores $G$, and trains new policies with the explored goals in order to expand $\Pi$. The explored goals and their corresponding policies are stored in a buffer $Z$ in the form of goal-policy pairs (i.e., $(g, \pi)$). The exploration strategy of GEP is designed to ensure that the policies in $\Pi$ are diverse enough such that they are able to be used for mapping with a wide range of human instructions in $C$. In this paper, a parameter space noise technique (Colas et al., 2018) is employed to enhance the diversity of goals explored by GEP, though other exploration techniques can be adopted for the same purpose. In the evaluation phase, the instruction-goal mapping module first maps a human instruction $c$ to its closest goal $g$ by iteratively searching for $g$ in $G$ using CEM, which employs $Dist_\phi(c, g)$ as the metric function for optimization. The implementation details of CEM is provided in our supplementary material. GSP then searches through the goal-policy pairs stored in $Z$, and outputs a policy $\pi$ that leads the robot to a final goal $g'$ which is the nearest neighbor of $g$. The implementation details of GSP is presented in Section 3.3.2.

### 3.2 INSTRUCTION-GOAL MAPPING MODULE

In this section, we provide the implementation details of the instruction-goal mapping module. We first walk through the functionality and architecture of the trajectory encoder. Then, we explain the training methodology of the distance function.

---

**Algorithm 1** Training procedure of $Dist_\phi(c, g)$

---

**Require:** Instruction Set $C$, Goal Set $G$, difference function $d$
**Ensure:** Learned Parameters with parameters $\phi$
 1: Initialize network weights $\phi$
 2: **while** not converged **do**
 3:     sample mini-batch (c, g) $\in (C, G)$
 4:     $L_\phi^{Dist} = |Dist_\phi(c, g) - d(\tau^c, \tau^g)|$
 5:     $\phi \leftarrow \phi - \eta \nabla_\phi L_\phi^{Dist}$
 6: **end while**

---

### 3.2.1 TRAJECTORY ENCODER

The objective of the trajectory encoder $f$ is to encode an arbitrary input trajectory $\tau$ into an n-dimensional representation $h \in \mathbb{R}^n$ that preserves the characteristics of $\tau$. Depending on the requirements of the tasks, $f$ can be implemented in different fashions, as long as the encoded $h$ is representative of $\tau$. In this paper, we implement two different trajectory encoders $f_{goal}$ and $f_{traj}$ for the goal-oriented tasks and trajectory-oriented tasks, respectively. The details of these two types of tasks are described in Section 4.1. For goal-oriented tasks, the representation $h$ is simply the final observation of $\tau$ (i.e., $o_T$ in Eq. (1)). For trajectory-oriented tasks, on the other hand, $h$ is a latent embedding encoded from $\tau$ by a auto-encoder based on a recurrent neural network (Sutskever et al., 2014). This design enables $f_{traj}$ to compress a complex $\tau$ performed by a robot's behavior policies $\pi$ into a simpler but feature-preserving representation format $h$, preventing the need to represent $\tau$ directly in terms of its sequences of observations $o_t$'s. The details of $f_{traj}$ is presented in our supplementary material.

### 3.2.2 DISTANCE FUNCTION

The distance function $Dist_\phi(c, g)$ serves as a metric function for measuring the similarity between any given pairs of $c$ and $g$. Instead of directly comparing the representations of $c$ and $g$, $Dist_\phi(c, g)$ is trained to predict the distance between $h_{\tau^c}$ and $h_{\tau^g}$ encoded from the corresponding $\tau^c$ and $\tau^g$ by the trajectory encoder $f_{traj}$. Similar approaches have been employed in the natural language processing (NLP) domain for constructing representations of sentences for years (Cho et al., 2014; Sutskever et al., 2014). Algorithm 1 outlines the pseudocode for the main steps of the distance function training procedure. The initialization takes place in line 1. A number of pairs of $(\tau^c, \tau^g)$ are first sampled from the training trajectories collected in advance, as described in Section 3.1. The distance $d$ of each sampled pair $(\tau^c, \tau^g)$ is then measured in terms of the $L^2$ distance of the latent embeddings ($h_{\tau^c}$, $h_{\tau^g}$), expressed as:

$$d(\tau^c, \tau^g) = \|h_{\tau^c} - h_{\tau^g}\|_2 . \tag{2}$$

The loss function $L^{Dist}$ of $Dist_\phi(c, g)$ is thus defined as:

$$L_\phi^{Dist} = |Dist_\phi(c, g) - d(\tau^c, \tau^g)|. \tag{3}$$

In this work, $Dist_\phi(c, g)$ is implemented as an one-layer fully-connected (FC) neural network. The parameters $\phi$ are iteratively updated such that the loss $L_\phi^{Dist}$ is minimized.

## 3.3 GOAL-POLICY MAPPING MODULE

In this section, we present the details of the goal-policy mapping module. We describe GEP and GSP separately in Sections 3.3.1 and 3.3.2, respectively.

### 3.3.1 GOAL EXPLORATION PROCESS

The GEP is employed to discover a set of policies corresponding to the specific goal as described in Sections 2.1 and 3.1. As the policy for reaching the goal is complex, it turns out to be intractable programming the policy manually. Moreover, supervised learning is impractical due to the expensive cost of the annotation. In contrast, GEP is able to discover the way to reach a variety of goals with a few prior knowledge of environment without labeling efforts. Therefore, we adopt GEP to alleviate the problems. In this paper, the goal space $G$ depends on the type of tasks, and the space of non-linear neural network utilized in GEP as a controller is defined by the policy space $\Pi$. In the bootstrap stage, we first sample a policy from $\Pi$ with uniform distribution $U$. On the other hand, in the random exploration stage, we first sample a goal $g$ from a specified goal space $G$. Then retrieve a

---

**Algorithm 2** Cross-entropy method

---

**Require:** Instruction $c$, distance function with parameters $\phi$
**Ensure:** Skill $z$
    Random sample skill $z^{(0)}$
2: Initial parameters mean and standard deviation $\mu^{(0)}, \sigma^{(0)}$
    Initial parameter N for the number of generating samples
4: Initial parameter Ne for the number of better samples to update $\mu, \sigma$
    Initial parameter maxits for the maximum iteration
6: $t \leftarrow 0$
    **while** not converge and $t <$ maxits **do**
8:    generate random sample $X = \{x_1, \cdots, x_N\}$ by Gaussian distribution based on $\mu^{(t)}, \sigma^{(t)}$
        $z^{(t+1)} = \arg\min_{x \in X} Dist_\phi(c, x)$
10:    $R = Dist_\phi(c, X)$
        sort $X$ by corresponding $R$ (in ascending order)
12:    $\mu^{(t+1)} = \text{mean}(X[1 : Ne])$
        $\sigma^{(t+1)} = \text{std}(X[1 : Ne])$
14:    $t = t + 1$
    **end while**
16: **return** $z^{(t)}$

---

closest policy $\pi$ searching by k-nearest neighbor, and perturb the selected policy with Gaussian noise $\mathcal{N}(0, \sigma^2)$.

### 3.3.2 GOAL SEARCHING PROCESS

In the goal searching process, we leverage the previous components, including $Dist_\phi(c, g)$ in the instruction-goal mapping module and the buffer $Z$. Given an instruction $c$, GSP first calculates the distances between $c$ and all the goals $g$'s stored in $Z$. The GSP then searches among the sets of goal-policy pairs with the minimum distance. In the case where the goal space $G$ is continuous and enumeration is not practical, a sampling-based searching algorithm is used instead to approximate the optimal solution that minimizes $Dist_\phi(c, g)$. Different from the enumeration method, the sampling-based searching algorithm seeks to find a satisfactory pair of mean and variance $(\mu_g, \sigma_g)$ to sample a normal distribution of $g$ within a given amount of time $T_S$. The sampling-based searching algorithm adopted in this paper is the cross entropy method (CEM), which is outlined in Algorithm 2 CEM casts the original searching problem into an optimization problem, which repeats the sampling procedure in lines [7-14] until either the convergence condition or $T_S$ is met. In each iteration, the pair $(\mu_g, \sigma_g)$ is updated toward the direction that decreases $Dist_\phi(c, g)$, which serves as the metric function for evaluating the distribution of the sampled $g$'s. Compared to stochastic gradient descent (SGD), CEM is a simple and directive-free algorithm for handling non-convex optimization problem in the continuous domain. A similar methodology has been employed and shown to be effective in [Kalashnikov et al.,2018] for online optimization.

## 4 EXPERIMENTAL RESULTS

In this section, we present the experimental results and discuss their implications. We start by a brief introduction to our experimental setup, which includes a description of the environment, the dataset preparation method, as well as the baseline models we used. Then, we compare the performances of the proposed approach against the baselines for both known and unknown instructions. Moreover, we demonstrate the adaptability of our modular approach by replacing the input instruction set to another. Finally, we provide a comprehensive ablation analysis of our approach.

### 4.1 EXPERIMENTAL SETUP

In this section, we present our experimental setups, including the environment and task settings, the baselines, as well as the training details of our method.

### 4.1.1 Environment Setting

We customize the Open-Ai gym environment[1] to evaluate the proposed methodology. Our experiments contain two kinds, 2D mass point, and Mujoco reacher. For the 2D mass point, an agent is placed at the lower part of a $2 \times 2$ unit lengths square arena surrounded by walls (barriers) at the beginning of an episode. During the episode, the agent always moves forward in fixed speed and rotate in ranging degree. The Action is one dimension space deciding the amount of rotation, and the observation for our agent getting from the environment contains two dimensions, x coordinate and y coordinate, to present its position. For a whole episode, the agent takes action every timestep and has at most 50 steps to explore the environment. While the Mujoco Reacher agent is initialized as a gesture pointing at the right side. A reacher consists of two movable fingers, one of its finger fixes at the center of a square arena, the other attaches on its first finger to stretch more further. A ball on the top of the second finger, called fingertip, is the part to decide where the Reacher pointing at. What reacher observes the word is its fingertip's position, coordinates in two dimension. The actions are two forces forcing vertically on its fingers respectively. By the maximum steps 15, the reacher takes action every step to move its fingertip everywhere.

With both the environments, we evaluate the proposed methodology in two different ways, goal-oriented task and trajectory-oriented task. Beginning with the goal-oriented task, for each episode, five different position targets are sampled (Chaplot et al., 2017) in the fixed position away from the agent with one to be randomly selected as an instructed target. An instruction for approaching one of the five targets in the arena is generated and passed to the agent at first. The instruction containing the symbols of the target object makes the agent be able to distinguish the correct one from the others. An episode terminates when the agent touches the objects, walls, or the maximum timesteps of the simulation is reached. To evaluate the performance of this task, we take the end position of the agent's trajectory to calculate the L2-norm by distance to instructed target position. Then comes to the trajectory-oriented task, there are only a few differences from above. For each episode, five separated position targets with two added targets appear in the environment. Within the instruction become approaching more than one targets, one of two added targets is selected as internal targets pair with one of the remaining targets. The evaluation is first to find the closest position to the instructed internal target in the whole trajectory, then find the other closest position to the instructed final target in the trajectory after above one. At last, we calculate the L2-norm by the two distance between agents' positions and instructed targets' position.

### 4.1.2 Task Settings

The proposed model is evaluated against the baselines in 2D point of mass environment and the locomotion control task in MuJoCo environment on two benchmark tasks: goal-oriented tasks and trajectory-oriented tasks. In the goal-oriented tasks, the robot is required to reach the goal specified in a given instruction. The instructions are represented in a 5D one-hot vectors that stand for targets in different positions. These tasks are directly and succinctly justify our proposed model being able to focus on a particular target. On the other hand, the trajectory-oriented tasks require the robot to not only reach the goal but also pass through a mid-point target. In other words, the robot 's objective is to extract the representation from the trajectory and be capable of telling the difference between different paths.

### 4.1.3 Baselines

In this paper, we adapt RL- and IL- based algorithm. In the RL training, we use proximal policy optimization proposed by OpenAI. Compare to general RL method, our model needs no reward signal received from the environment, and it preserves the hard work of tuning the reward fucntion for ours.

In the imitation learning baselines, we adapt the Behavior Cloning (BC) approach. In the phase of collecting expert data, we obtain the demonstration either from human or well-pretrained RL model. Then agent learns to fit to the demonstraion. BC performs well in small environment, while getting an unpleasant result if agent is unable to explore most of states of the enviroment through the expert data.

---

[1]https://gym.openai.com/

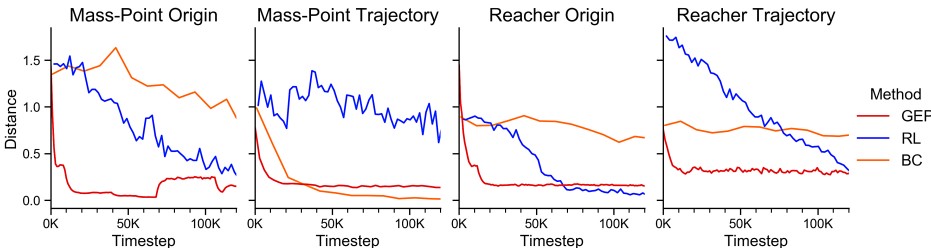

**Figure 2: Data-efficiency comparison of goal-oriented tasks and trajectory-oriented tasks**

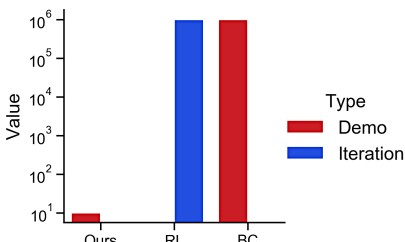

**Figure 3: Overview of the proposed framework.**

### 4.1.4 TRAINING DETAIL OF OUR METHOD

The detail configurations of our method are presented in this section. First, the settings of goal space are described. Next, the format of the human demonstrations are explained. Finally, the source of human demonstrations for distance function training is presented. The settings of goal space vary in goal-oriented and trajectory-oriented tasks. First, the goal space of goal-oriented task is defined by a 2-dimensional continuous space in the 2-D mass point and Reacher environments. Each element in the goal space indicates a coordinate in the environment. Secondly, a 4-dimensional continuous space represents the goal space for trajectory-oriented task. The former two dimensions represent the coordinate of an intermediate point while the latter two indicate the target point in both envir

### 4.2 COMPARISON OF DATA-EFFICIENCY TOWARDS A FIXED INSTRUCTION SET AND A FIXED ACTION SPACE

Here we first compare the performance of our method with baseline approaches by evaluating the data-efficiency towards a fixed instruction set and a fixed action space. From Fig. 2 Mass-Point origin and Reacher origin. For both environments, we found that our method fits the task quicker than $RL$ and $BC$ approaches. In the Mass-Point task, our approach even performs the best as over tenth thousand timesteps, while in Reacher task, our method drop the loss fast and stay at lower and stable phase. The $RL$ approach needs more than fifty thousand timesteps to perform better, with the $BC$ approaches never drops that low in ten thousand timesteps. Otherwise, we found that our method bouncing a little bit after fifty thousand timesteps in Mass-Point task. Due to our method is not a learning module, sometimes it may find other unseen situation to previous experiences to influence the predicting preference.

### 4.3 COMPARISON OF DATA-EFFICIENCY TOWARDS AN UNFAMILIAR INSTRUCTION SET AND A FIXED ACTION SPACE

Fig. 3 plots the data requirement for all the methods towards an unfamiliar instruction set and a fixed action space. In all of the tasks, our method yields superior data-efficiency to the baselines. In $RL$, it can be seen that it require a large amount of iterations to adapt to an unfamiliar instruction set. On the other hand, the $BC$ turns out to have similar result as $RL$. However, the $BC$ require human demonstrations while $RL$ requires the re-trained or even re-designed reward function. Our proposed method shows the data-efficiency due to the distance function, wherein such component in our model benefits from mapping unfamiliar data set with limited human demonstration.

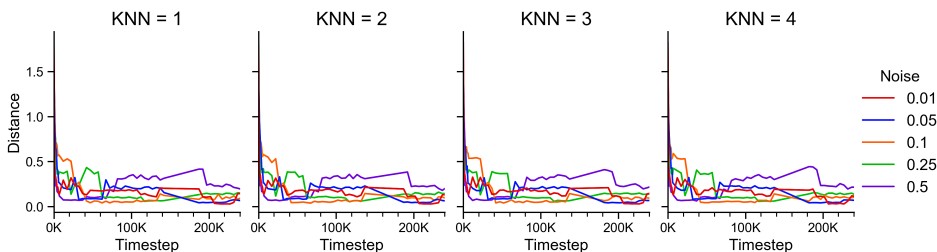

**Figure 4: Overview of the proposed framework.**

## 4.4 COMPARISON OF DATA-EFFICIENCY TOWARDS A FIXED INSTRUCTION SET AND A NEW ACTION SPACE

For this part, we compare the performance of our method with baseline approaches by evaluating the data-efficiency towards a fixed instruction set and a new action space. In this experiment, we change the action space and evaluate the performance of both environments. Our method only explores the environment with new action space but not training the RAE and distance function. For the $RL$ and $BC$ approaches, they need to train their module from very beginning to fit the experiment. Moreover, refers to the experiment results from Fig. 4, our method still drops the loss fast to prove our data-efficiency ability with adaptability.

## 4.5 ABLATIVE STUDIES ON SCALE OF NOISE IN GEP

From Fig. 4, it can be observed that different noises adding in the random exploration stage result in diverse converge rate. Using $noise = 0.5$, which is the maximum in our experiment, decreases the most fast, however, it bound back to a high distance after around 100K. On the other hand, though adding $noise = 0.01$ as the minimum turns out to decrease much slower than previous choice does not re-bounce. It shows that though lager noises can fasten the goals been explored diversely, it tends to leads to an unstable situation. Finally, adding $noise = 0.1$ is the most robust choice, while it explores the new goals straight and narrow.

## 5 CONCLUSION

Wee presented a modular approach for separating the instruction-to-action mapping procedure into two separate stages. The first stage maps an input instruction to a goal, while the second stage maps the goal to an appropriate policy selected from a set of robot policies. We implemented the above two stages as a framework consisting of an instruction-goal mapping module and a goal-policy mapping module. Our experimental results show that the proposed method is able to learn an effective instruction-to-action mapping procedure in an environment with a given instruction set more efficiently than the baselines.

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
