# OpenReview forum: "A Self-Supervised Method for Mapping Human Instructions to Robot Policies"
_ICLR.cc/2019/Conference_

### Official Review · AnonReviewer2 · 2018-10-31

**Rating:** 2
**Confidence:** 4

**Review:**

This paper presents an instruction-following model consisting of two modules: a
goal-prediction model that maps commands to goal representations, and an
execution model that maps goal representations to policies. The second module is
trained without command supervision via a goal exploration process, while the
first module is trained supervisedly in a metric learning framework.

This paper contains an important core insight---much of what's hard about
instruction following is generic planning behavior that doesn't depend on the
semantics of instructions, and pre-learning this behavior makes it possible to
use natural language supervision more effectively. However, the paper also
contains a number of serious evaluation and presentation issues. It is obviously
not ready to publish (uncaptioned figures, paragraphs interrupted mid-sentence,
etc.) and should not have been submitted to ICLR in its present form.

SUPERVISION AND COMPARISONS

I found comparisons between supervision conditions in this paper difficult to
understand. It is claimed that the natural language instruction following
approaches described in the first paragraph "require a large amount of human
supervision" in the form of action sequences. This is not exactly true, as some
approaches (e.g. Artzi 2013), can be trained with only task completion signals.
More problematically, all these approaches are contrasted with reinforcement and
imitation learning approaches, which are claimed to use "little human
supervision". In fact, most of the approaches listed in this section use exactly
the same supervision---either action sequences (imitation learning) or task
completion signals (reinforcement learning). Indeed, the primary distinction is
that the "NLP-style" approaches are typically evaluated on their ability to
generalize to new instructions, while the "RL-style" approaches are evaluated on
the (easier) problem of fitting the complete instruction distribution as quickly
as possible.

This confusion carries into the evaluation of the approach proposed in this
paper, which is compared to RL and IL baselines. It's hard to tell from the
text, but it appears that this is an "RL-style" evaluation setting, where we
only care about rapid convergence rather than generalization. But the baselines
are inadequately described, and it's not clear to me that they condition on the
commands at all. More significantly, it's not clear what an evaluation based on
"timesteps" means for a behavior-cloning approach---is this the number of
distinct trajectories observed? The number of gradient steps taken? Without
these explanations it is impossible to interpret the experimental results.

GENERALITY OF PROPOSED APPROACH

Despite the advantages of the high-level two-phase model proposed, the specific
implementation in this paper has two significant shortcomings:

- No evidence that it works with real language: despite numerous claims
  throughout the paper that the model is designed to interpret "human
  instructions", it is revealed on p7 that these instructions consist of one or two
  5-way indicator features. This is an extremely impoverished instruction space,
  especially compared to the numerous papers cited in the introduction that make
  use of large datasets of complex natural-language strings generated by human
  annotators. The present experiments do not support the use of the word "human"
  anywhere in the paper.

- No support for combinatorial action spaces. Even if we set aside the
  distinctions between human-generated instructions and synthetic command
  languages like used in Hermann Hill & al., the goal -> policy module is
  defined by a buffer of cached trajectories and goal representations. While
  this works for the simple environments considered in this paper, it cannot
  generalize to real-world instruction-following scenarios where the number of
  distinct goal configurations is too large to tractably enumerate. Again, this
  is a shortcoming that existing approaches do not suffer from (given
  appropriate assumptions about the structure of goal space), so the lack of
  comparisons is problematic.

CLARITY

The whole paper would benefit from copy-editing by an experienced English
speaker, but a few sections are particularly problematic:

- The first paragraph of 4.1.1 is extremely difficult to understand What does
  the fingertip do? What exactly is the action space?

- The end of the second paragraph is also difficult to understand; after reading
  it I still don't know what the extra "position" targets do.

- 4.1.4 is cut off mid-way through a sentence.

- last sentence of 4.2

The figures are also impossible to interpret: three of the four are captioned
"overview of the proposed framework", and none are titled.

---

### Official Review · AnonReviewer1 · 2018-11-02
**Proposed method has several limitations, experimental setup is unclear and the results are not convincing.**

**Rating:** 3
**Confidence:** 5

**Review:**

This submission proposes a method for learning to follow instructions by splitting the policy into two stages: human instructions to robot-interpretable goals and goals to actions. The authors claim to achieve better data efficiency, adaptability, and generalization as compared to the baselines.

Here are some comments/questions:
- One of the biggest limitations of the proposed method is that it can only work for one-to-one or many-to-one mapping of instructions to goals. As I understand (please correct me if I am wrong), the method can not work for contextual instructions where the goal depends on the environment and the same instruction can map to different goals, such as 'Go to the largest/farthest object'.
- Another limitation of the method is that it requires a set of goals G, which is not trivial to obtain especially in partially observable environments such as embodied navigation in 3D space.
- The experimental setup is unclear and several crucial details are missing:
	- "An instruction for approaching one of the five targets in the arena is generated and passed to the agent at first." -> how is the instruction generated?
	- There's no example of the environment or the instruction in the submission
	- "Within the instruction become approaching more than one targets, one of two added targets is selected as internal targets pair with one of the remaining targets." I do not understand this sentence. How are the targets generated in the trajectory-oriented task? How are the instructions generated in this task?
- Experimental results are not convincing:
	- The introduction motivates the need for understanding human instructions and the abstract says 'Given a human instruction', but I believe experiments do not have any human instructions.
	- All the environments seem to be fully-observable, it is not clear whether the method would work in partially-observable environments.
	- Only vanilla PPO and BC cloning are used as baselines. There are several competing methods for following instructions which the authors cite such as Hermann et al. 2017, Chaplot et al. 2017, Misra et al. 2017, etc. Why weren't any of these approaches used as a baseline?
- The submission requires proof-reading, there are several typos in the manuscript (some are listed below), some of them make it very difficult to understand the setting.

- Typos:
- Sec 3.1 on Pg 4 mentions 'CEM' multiple times, it's not defined until 3.3.2 on Pg 6.
- Pg 3 Theses sets -> These sets
- Pg 7 where the Reacher pointing at -> where the Reacher is pointing at
- Pg 7 What reacher observes the word is its fingertip’s position, coordinates in two dimension. -> something is wrong in this sentence.
- Pg 7 Then comes to the trajectory-oriented task, there are only a few differences from above -> something is wrong in this sentence.
- Pg 7 Within the instruction become approaching more than one targets -> something is wrong here

---

### Official Review · AnonReviewer3 · 2018-11-05
**Overall idea is interesting, but novelty is limited and evaluation is poor**

**Rating:** 4
**Confidence:** 5

**Review:**

The paper proposes a modular approach to the problem of mapping instructions to robot actions. The first of two modules is responsible for learning a goal embedding of a given instruction using a learned distance function. The second module is responsible for mapping goals from this embedding space to control policies. Such a modular approach has the advantage that the instruction-to-goal and goal-to-policy mappings can be trained separately and, in principle, allow for swapping in different modules. The paper evaluates the method in various simulated domains and compares against RL and IL baselines.

STRENGTHS

+ Decoupling instruction-to-action mapping by introducing goals as a learned intermediate representation has advantages, particularly for goal-directed instructions. Notably, these together with the ability to train the components separately will generally increase the efficiency of learning.


WEAKNESSES

- The algorithmic contribution is relatively minor, while the technical merits of the approach are questionable.

- The goal-policy mapping approach would presumably restrict the robot to goals experienced during training, preventing generalization to new goals. This is in contrast to semantic parsing and symbol grounding models, which exploit the compositionality of language to generalize to new instructions.

- The trajectory encoder operates differently for goal-oriented vs. trajectory-oriented instructions, however it is not clear how a given instruction is identified as being goal- vs. trajectory-oriented.

- While there are advantages to training the modules separately, there is a risk that they are reasoning over different portions of the goal space.

- A contrastive loss would seemingly be more appropriate for learning the instruction-goal distance function.

- The goal search process relies on a number of user-defined parameters

- The nature of the instructions used for experimental evaluations is unclear. Are they free-form instructions? How many are there? Where do they come from? How different are the familiar and unfamiliar instructions?

- Similarly, what is the nature of the different action spaces?

- The domains considered for experimental evaluation are particularly simple. It would be better to evaluate on one of the few common benchmarks for robot language understanding, e.g., the SAIL corpus, which considers trajectory-oriented instructions.

- The paper provides insufficient details regarding the RL and IL baselines, making it impossible to judge their merits.

- The paper initially states that this distance function is computed from learned embeddings of human demonstrations, however these are presumably instructions rather than demonstrations.

- I wouldn't consider the results reported in Section 4.5 to be ablative studies.

- The paper incorrectly references Mei et al. 2016 when stating that methods require a large amount of human supervision (data annotation) and/or linguistic knowledge. In fact Mei et al. 2016 requires no human annotation or linguistic knowledge.

- Relevant to the discussion of learning from demonstration for language understanding is the following paper by Duvallet et al.

Duvalet, Kollar, and Stentz, "Imitation learning for natural language direction following through unknown environments," ICRA 2014

- The paper is overly verbose and redundant in places.

- There are several grammatical errors

- The captions for Figures 3 and 4 are copied from Figure 1.

---

### Meta-Review · Area_Chair1 · 2018-12-12
**nice, but unripe**

**Confidence:** 5
**Recommendation:** Reject

**Metareview:**

The paper proposes a novel approach to interfacing robots with humans, or rather vv: by mapping instructions to goals, and goals to robot actions.   A possibly nice idea, and possibly good for more efficient learning.

But the technical realisation is less strong than the initial idea.  The original idea merits a good evaluation, and the authors are strongly encouraged to follow up on this idea and realise it, towards a stronger publication.

It be noted that the authors refrained from using the rebuttal phase.